# Exploring Sentence Vector Spaces through Automatic Summarization

## Abstract

Vector semantics, especially sentence vectors, have recently been successfully in many areas of natural language processing. However, relatively little work has explored the internal structure and properties of spaces of sentence vectors. In this paper, we will explore the properties of sentence vectors by studying a particular real-world application: Automatic Summarization. In particular, we show that cosine similarity between sentence vectors and document vectors is strongly correlated with sentence importance and that vector semantics can identify and correct gaps between the sentences chosen so far and the document. In addition, we identify specific dimensions which are linked to effective summaries. To our knowledge, this is the first time specific dimensions of sentence embeddings have been connected to sentence properties. We also compare the features of different methods of sentence embeddings. Many of these insights have applications in uses of sentence embeddings far beyond summarization.

## 1 Introduction

*Vector semantics* have been growing in popularity for many other natural language processing applications. Vector semantics attempt to represent words as vectors in a high-dimensional space, where vectors which are close to each other have similar meanings. Various models of vector semantics have been proposed, such as LSA (Landauer & Dumais, 1997), word2vec (Mikolov et al., 2013), and GLOVE(Pennington et al., 2014), and these models have proved to be successful in other natural language processing applications.

While these models work well for individual words, producing equivalent vectors for sentences or documents has proven to be more difficult.

In recent years, a number of techniques for sentence embeddings have emerged. One promising method is *paragraph vectors* (Also known as Doc2Vec), described by Le and Mikolov Le & Mikolov (2014). The model behind paragraph vectors resembles that behind word2vec, except that a classifier uses an additional 'paragraph vector' to predict words in a *Skip-Gram* model.

Another model, *skip-thoughts*, attempts to extend the word2vec model in a different way (Kiros et al., 2015). The center of the skip-thought model is an encoder-decoder neural network. The result, *skip-thought vectors*, achieve good performance on a wide variety of natural language tasks.

Simpler approaches based on linear combinations of the word vectors have managed to achieve state-of-the-art results for non-domain-specific tasks (Wieting et al., 2015). Arora et al.Arora et al. (2016) offer one particularly promising such approach, which was found to achieve equal or greater performance in some tasks than more complicated supervised learning methods.

## 2 Problem Description

The large volume of news articles published every day motivates effective forms of automatic summarization. The most basic and most well-studied form of automatic summarization is sentence extraction, in which entire unmodified sentences are selected from the original document. These selected sentences are concatenated to form a short summary, which ideally contains the most important information from the original document while avoiding redundancy.

Any practical summarization system would likely include other steps after sentence selection, such as *sentence reordering* or *text-simplification*. However, effective algorithms for these tasks already exist, and we are primarily concerned with sentence selection in this paper.

We are primarily concerned with multi-document summarization algorithms.

The nature of summarization requires some specific properties from sentence vectors, making it a practical real world example for which current sentence embedding methods can be easily used. In particular, news summarization may not require some of the nuances in meaning to be represented, as we are primarily concerned with the topic, not the meaning, of a sentence. In particular, news articles will rarely contain sentences expressing contradictory views on the same subject. For instance, our algorithm will rarely need to differentiate between the sentence embeddings for sentences such as "John loves Mary" and "Mary loves John", which have divergent meanings but the same words. This is in sharp contrast to typical testing cases for sentence embeddings, such as the detection of paraphrased sentences. Then, summarization gives us an opportunity to compare the effectiveness of different sentence embeddings in practice.

## 3 RELATED WORK

### 3.1 EXTRACTIVE SUMMARIZATION

State-of-the-art extractive summarization techniques have been achieved with a wide variety of methods. Here, we provide a short overview of some recent successful techniques techniques.

Cao et al. Cao et al. (2015) used a recursive neural network, which operates on a parsing tree of a sentence, to rank sentences for summarization.

Cheng and Lapata Cheng & Lapata (2016) successfully used a neural-network-based sentence extractor, which considered the document encodings and the previously selected sentences, as well as the current sentence in making its decisions.

Parveen et al. Parveen & Strube (2015) used a graph-based approach, modeling a document as "a bipartite graph consisting of sentence and entity nodes".

Ren et al.Ren et al. achieved state-of-the-art results through a regression-based approach. A variety of engineered features are used as inputs into a regression model. The single highest rated sentence is selected, and the process is repeated.

### 3.2 PREVIOUS EMBEDDING-BASED APPROACHES

To our knowledge, no one has explored the use of modern sentence embedding methods, such as Paragraph Vectors or Skip-Thought vectors, in summarization. However, some work has been done on summarization using word2vec representations.

Gong and Liu Gong & Liu (2001) presented a version of text summarization based on vector semantics. However, instead of using the embeddings of a word as determined by a larger corpus, they attempted to calculate the embedding of a word based off of analysis only on the document in question. In addition they used LSA instead of newer techniques such as word2vec.

Kageback et al. Kågebäck et al. (2014) used cosine similarity between the sum of word2vec embeddings, as well as a recursive auto-encoder, to modify another standard summarization algorithm (sub-modular optimization)

Ren et al. Ren et al. used "Average Word Embedding" as one of many independent variables in a regression model, but the degree to which word embeddings effected the final model was not clear.

Cheng and Lapata Cheng & Lapata (2016) used word embeddings as the input to a neural network as part of summarization, but they did not directly compare embeddings. They used a single-layer convolutional neural network to encode sentences, and a LSTM neural network to encode documents.

Nayeem and Chali Nayeem & Chali (2017) recently achieved state-of-the-art results using a modified version of LexRank, using a combination of cosine similarity of weighted averages of vectors and named entity overlap.

## 4 METHODS

To explore potential sentence embeddings, we implement the four sentence embeddings above as *vector functions*, which convert sentences or documents to vectors.

### 4.1 VECTOR FUNCTIONS

- *SIF Average*: The most basic sentence embedding is simply the weighted average of word vectors from Arora et al.Arora et al. (2016), without the common component removal. We use the Brown corpus (Francis & Kucera, 1979) for word frequency information.

- *Arora*: This method is simply the method described in Arora et al. It is equivalent to the one above, except with common component removal added. We use the Brown corpus both to compute the common component vector, and for word frequency information.

- *Paragraph Vectors*: The paragraph vector approach described above. We used the 300-dimensional DBOW model pretrained by Lau et al.Lau & Baldwin (2016) on the wikipedia corpus.

- *Skip-Thought Vectors* The skip-thought vector approach described above. We used the 4800-dimensional combined-skip model (Kiros et al., 2015; Al-Rfou et al., 2016).

All sentence embeddings are normalized to produce unit vectors

### 4.2 POTENTIAL SELECTOR FUNCTIONS

To explore the design space, we consider combinations of *vector functions* and *selector functions*, functions which, given vector representations for sentences, extracts a summary. We present a large variety of example selector functions in order to allow us to explore interaction effects between selector functions and vector functions.

- *Near*: The most basic selector function, *Near*, selects the sentences whose sentence vectors have the highest cosine similarity with the document vector

- *Near Nonredundant*: An attempt at balancing redundancy with salience, Near Nonredundant down-weights the cosine similarity scores by their average cosine similarity the sentences selected so far. Because this redundancy measure is strongly (quadratically) correlated with cosine similarity to the document, we fit a regression for redundancy for each vector function, and use the residual on this regression for the final algorithm.

- *LexRank*: Our next selector is based off of the classical LexRank algorithm (Erkan & Radev, 2004). We use a modified version of this algorithm, where the weights of edges are determined by the cosine similarity between sentence embeddings.

- *Cluster*: We use an Agglomerative Clustering algorithm (using cosine similarity as its distance metric) to find clusters in the set of sentence embeddings. We then find the sentence closest to the average of each cluster and add it to the summary. To ensure we find summaries which meet the word-length requirement, we increase the number of clusters we search for until we have selected sentences totaling 100 words.

- *Greedy*: The greedy selector, at each step, selects the sentence such that the cosine similarity of the new summary (including previously selected sentences) is maximized. This is subtly different than the Near selector for average-based vector functions, but significantly different for Paragraph Vectors.

- *Brute Force*: Another attempt at optimizing the cosine similarity between the summary and the document, this selector creates a pool of the 20 sentences with the highest cosine similarity. From this pool, every combination of sentences (with an appropriate word count) is tried, and the combination with the highest cosine similarity is selected as the summary.

- *Max Similarity*: A proof-of-concept selector which computes results for both the Greedy and Brute Force selectors and then selects the result with the highest cosine similarity to the document vector.

- *Near-then-Redundancy*: Starting with the pool from the Brute Force algorithm, this algorithm optimizes via brute force to minimize redundancy (defined as the average cosine similarity between pairs of sentences). Note that the size of the sentence pool, which is essentially a computational shortcut in the Brute Force selector, is now a performance-critical hyper-parameter.

- *PCA*: This selector performs Principal Component Analysis (PCA) on the set of sentence vectors in a document. Then, the algorithm selects the one sentence closest to the first component, one sentence closest to the second component, and so on, until the length capacity is met.

- Random: This selector simply selects sentences at random, until the word limit is reached. This provides a lower-bound on the performance of an effective algorithm, and is used for baseline comparisons

## 5 PERFORMANCE OF SELECTOR FUNCTIONS

|                      | SIF Average | Arora       | Paragraph Vectors | Skipthought  |
|----------------------|-------------|-------------|-------------------|--------------|
| **LexRank**          | 32.6 (6.8)  | 32.6 (6.8)  | 32.6 (6.8)        | **32.6 (6.8)** |
| **Near Nonredundant** | 33.6 (6.1)  | **34.5** (6.3) | 32.6 (5.5)     | 32.1 (4.9)   |
| **Brute Force**      | 32.0 (5.7)  | 32.2 (6.3)  | **33.0 (6.6)**    | 31.4 (4.5)   |
| **Near-then-Redundancy** | 33.2 (6.2) | 34.2 **(6.9)** | 31.5 (5.4)   | 33.1(5.3)    |
| **PCA**              | 32.9 (5.6)  | 33.5 (5.6)  | 32.0 (5.5)        | NA           |
| **Max Similarity**   | 32.0 (5.7)  | 32.2 (6.3)  | **33.0 (6.6)**    | NA           |
| **Greedy**           | **35.1 (7.0)** | 33.1 (6.0) | NA              | NA           |
| **Near**             | 32.5 (5.4)  | 32.2 (5.5)  | 33.1 (6.1)        | NA           |
| **Cluster**          | NA          | NA          | NA                | 32.1 (4.6)   |

Table 1: ROUGE-1 Results on the DUC 2004 dataset. ROUGE-2 results in parentheses. All combinations which do not perform significantly better than random chance ($p < .05$, using a paired t-test) are replaced with 'NA' for clarity. SIF Average with either Max Similarity or Brute Force were included, despite having p=.051. In addition, one combination (Max Similarity with Skipthought Vectors) are not computed, but are not expected to perform better than chance. Selector Functions are roughly organized according to the vector functions with which they are effective. For Skipthought vectors, *docvec-avg* is used (Section 6.5)

### 5.1 EXPERIMENTAL EVALUATION

Because evaluation of summarization is fundamentally a subjective task, human evaluations are ideal. However, human evaluations are often expensive and time-consuming to obtain. Luckily, some metrics of automatically evaluating summaries, by comparison to a human-written summary, have been developed. Traditionally, various forms of the ROUGE metric, which compare shared n-grams, have been used.(Lin, 2004). ROUGE has been shown to correlate strongly with human judgments(Rankel et al., 2013), and is our primary metric for evaluating summaries[1]. We report ROUGE-1 and ROUGE-2 statistics, which correspond to unigrams and bigrams, respectively.

We split the document clusters in the DUC 2004 dataset into a testing set and a validation set of approximately equal sizes. The pre-defined training set of the DUC 2001 dataset was used as a training set for some of the graphs and data analysis presented here.

### 5.2 RESULTS

We present results for Multi-Document Summarization on the DUC 2004 dataset (Table 1). A few notes on the results:

- The best performing selector, Greedy, is both very simple and based on fundamental principles of vector semantics.

---

[1]We trucate summaries to 100 words and use the following parameters, for direct comparison with Hong et al. Hong et al. (2014): -n 4 -m -a -l 100 -x -c 95 -r 1000 -f A -p 0.5 -t 0.

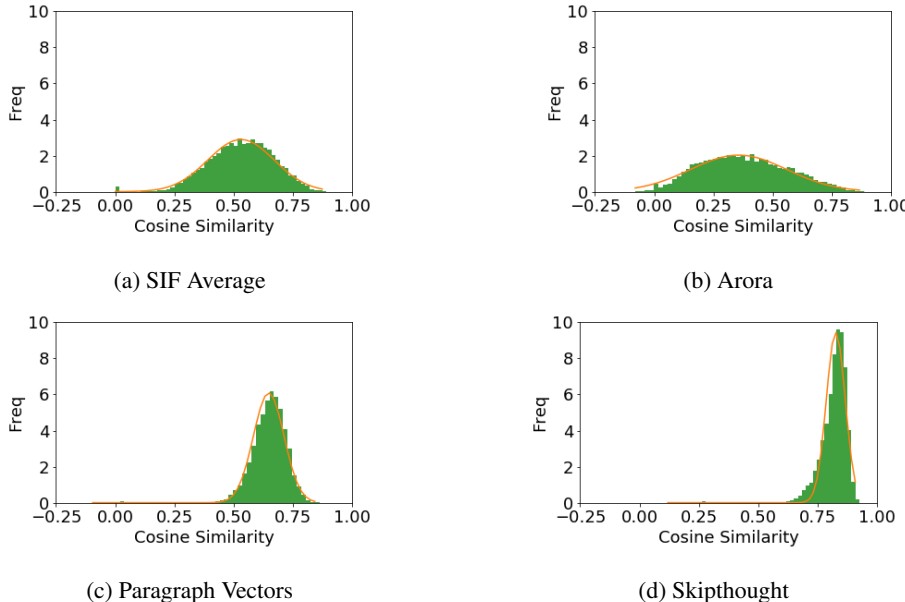

Figure 1: Distribution of cosine similarity scores between each sentence vector and their corresponding document vector, for all four vector functions.

- Paragraph Vectors work much worse with the Clustering and Greedy algorithms, and work much better with Near and SVMs.

- Many combinations of selector function and vector function do not work above the level of random chance.

- In general, despite their sophistication, Paragraph Vectors and Skip-Thought vectors perform worse than much more basic approaches.

# 6 DISCUSSION

Despite the poor performance of our models compared to the baselines, analyses of the underlying data provide many useful insights into the behavior of vector semantics in real-world tasks.

## 6.1 DISTRIBUTIONS OF COSINE SCORES

The cosine scores between all sentence vectors and the corresponding document vectors follow a normal distribution for all vector functions (Fig. 1), but this effect is most pronounced for paragraph vectors ($r^2 = .996$). In addition, the sentence embeddings for paragraph vectors and skip-thought vectors are far closer to the document embedding than would be expected from a random distribution, with mean cosine similarities of .65 and .84, respectively (Unsurprisingly, this also holds for Average and Arora vectors).

## 6.2 CORRELATION OF COSINE SCORES WITH GOOD SUMMARIES

By identifying the sentences present in an optimal summarization, we show that optimal sentences have higher cosine scores, and that this effect is increased after adjusting cosine scores for word length (Fig. 2). However, there is a lot of overlap, implying that, although this method has some power to discern good summaries from bad summaries, the power of this method alone is not high enough to product good summaries.

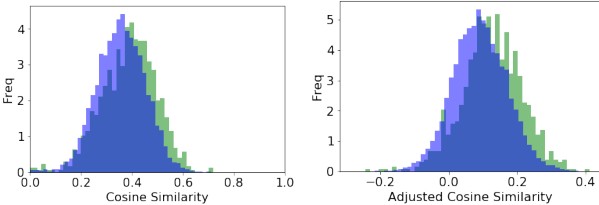

Figure 2: Cosine Similarity to the Document Vector for non-optimal (blue) and optimal (green) sentences. Figure on the right shows Cosine Similarity adjusted for sentence word count.

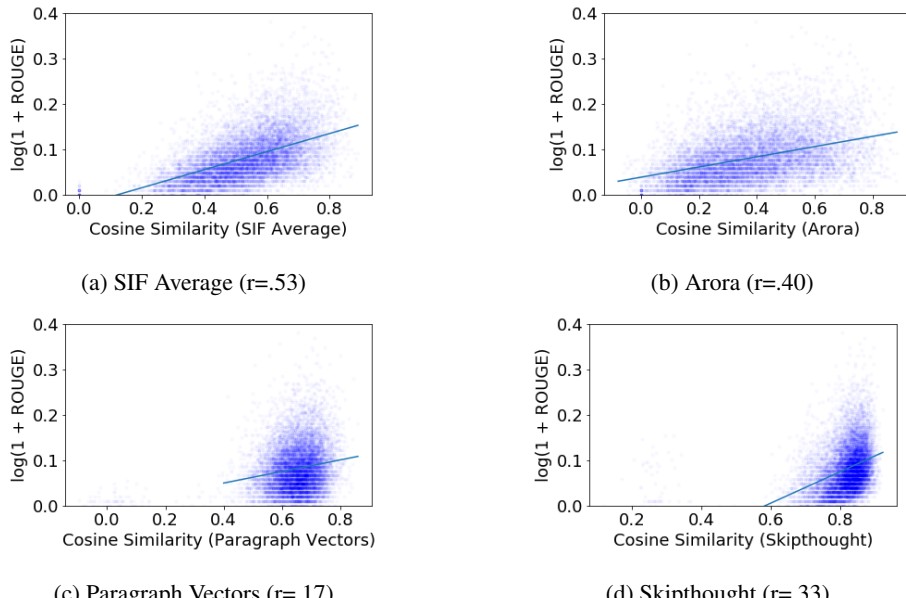

Figure 3: Correlation of ROUGE scores and Cosine Similarity scores per sentence. ROUGE scores transformed with $r' = log(1 + r)$, to account for zero values. Some high-leverage points have been excluded for Paragraph Vectors and Skipthought.

## 6.3 REGRESSION ON VECTOR DIMENSIONS

We calculated the isolated ROUGE score of each individual sentence in the training set, and the sentence embeddings for these sentences on all four vector functions. To partially eliminate the effects of the sentence's context, we subtract the corresponding document vector from all sentence vectors before regression.

Due to the large number of predictor variables, we use a Bonferroni correction, considering values significant only if they have p-values of $\frac{\alpha}{n}$, which, for $\alpha = .05$, corresponds approximately to $p < .00001$ for the skip-thought vectors, and $p < .00016$ for all other vectors.

Three dimensions are significant at this level for *SIF Average* vectors. No dimensions are significant at this level for *Arora* vectors, though the three values significant for *SIF Average* achieve p values of .0021, .0485 and .0006. 29 dimensions are significant for *Paragraph Vectors*. 5 dimensions are significant, at the much higher threshold, for *Skip-Thought Vectors*. It appears that these specific dimensions correspond to aspects of a sentence that make it somehow more suited for a summary. Despite the theoretical implications of this result, the regression models do not have enough predictive power to create good summaries by themselves.

## 6.4 THE PERFORMANCE OF THE GREEDY ALGORITHM

The Greedy algorithm is the most successful algorithm among those we have explored. As its name implies, the Greedy algorithm appears to be simply an attempt at maximizing the following objective function:

$$f_{cos}(summary) = vector(summary) \cdot vector(document) \tag{1}$$

Of course, any simple objective function can only be an approximation to the informally-defined criteria for good summaries. Even so, Table 1 suggests that the performance of the greedy algorithm is not based on the accuracy of the corresponding objective function. In particular, the two other strategies which try to maximize the same objective functio (Brute force, and Maximum Similarity) consistently and significantly create summaries with higher cosine similarity to the document, outperforming the Greedy selector on its objective function, but both of these algorithms perform much worse than the Greedy algorithm.

Deeper analysis into the decisions of the Greedy algorithm reveals some reasons for this discrepancy. It appears that the good performance of the Greedy algorithm results not from the associated objective function, but by the way in which it maximizes this objective function. In particular, the Greedy algorithm selects sentences with low cosine similarity scores in a vacuum, but which increase the cosine similarity of the overall sentence.

To understand why this is true, it we consider the step-by-step behavior of the Greedy algorithm, which can be understood as maximizing the following equation (derived in Appendix A):

$$\bar{s}_{i+1} = \operatorname*{argmax}_{\bar{s} \in S'} \frac{i(\bar{d} \cdot \bar{s}_p) + \bar{d} \cdot \bar{s}}{\sqrt{i^2 + 1 + 2i\bar{s}_p \cdot \bar{s}}} \tag{2}$$

$$\text{where } \bar{s}_p = \frac{\sum_{j=0}^{i} \bar{s}_j}{\left\| \sum_{j=0}^{i} \bar{s}_j \right\|}$$

Note that this equation consists of three parts: $i\bar{d} \cdot \bar{s}_1$ (a constant wrt. $\bar{s}$), $\bar{d} \cdot \bar{s}$, which is simply the salience of a sentence measured by cosine similarity, and the denominator, which is essentially a measure of redundancy. Not only does this simple metric lead to a 'natural' penalty for redundancy, it performs better than our handcrafted redundancy penalties.

The way this expression scales when picking the $i^{th}$ sentence is particularity noteworthy: the behavior of this function changes as $i$ increases. In particular, the function becomes more sensitive to redundancy, and less sensitive to salience, as the algorithm selects more sentences. In other words, the algorithm will first try to select important sentences, and then select sentences to fill in the gaps. This result, and the success of the resulting algorithm, has implications for balancing salience and redundancy in future summarization algorithms.

## 6.5 DOCUMENT VECTOR COMPUTATION

In general, there are two ways to compute a document vector. The most obvious is to pass the entire text of the document into the vector function. This has two theoretical problems. The first is that the 'documents' in our algorithms are really clusters of documents, and are therefore non-coherent. The second is that Skip-thought vectors are not designed to handle text longer than a sentence. However, an alternative document vector, *docvec-avg*, is defined as the mean of the (normalized) sentence vectors. This corresponds to treating the document as a collection of sentences, instead of a collection of words. We compare the two results here, and present full results in Appendix B.

As expected, Skipthought vectors, which are not designed for text larger than a sentence, perform significantly better with the *docvec-avg* strategy. More notable is the poor performance of the *docvec-avg* strategy with Paragraph Vectors. The size of the performance gap here implies that Paragraph Vectors can combine information from multiple sentences in a manner more sophisticated than simple averaging.

More interesting is the performance for SIF Average and Arora vectors. For these vector functions, which are based on taking the average of words, *docvec-avg* very closely resembles the simple

strategy. And yet there is a small but significant performance gap. The difference between the two document vectors is the weighting. *Docvec-avg*, which normalizes vectors before adding them together, removes some weighting information present in the simple strategy. In particular, the simple strategy assigns more weight to sentences with a lot of highly-weighted words. Presumably, *docvec-avg*, by ignoring this weighting, leaves out useful information. This hypothesis is supported by the greater performance gap for Arora vectors, which effectively downweights certain common words and therefore could be expected to carry more information in word weightings. Similar, but much smaller, gaps exist when computing the vectors for summaries at each step in the greedy algorithm.

### 6.6 PROPERTIES OF DIFFERENT SENTENCE EMBEDDINGS

We present a broad comparison of the properties of different sentence embedding schemes.

#### 6.6.1 SIF AVERAGE/ARORA VECTORS

Three selector functions perform better with both SIF Average and Arora vectors: Near Nonredundant, Greedy, and PCA. These functions seem to be unified by their comparisons between the vectors of sentence embeddings (implicitly, in the case of the greedy algorithm). These selector functions correspond to the most basic test for sentence embeddings: Judging the similarity of two sentences.

The exact difference the common component removal makes is less clear. Arora vectors hold a slight performance edge for all selectors except for Near and Greedy (the Greedy algorithm loses a full 2 points).

#### 6.6.2 PARAGRAPH VECTORS

Two selector functions perform better with Paragraph Vectors: Near and Brute Force. Both of these are very similar: They require comparing sentence vectors to the document vector. The poor performance on algorithms such as Near Nonredundant suggests that Paragraph Vectors are especially poor at comparing sentence vectors to each other. These results suggest that Paragraph Vectors are especially good at computing document vectors, a hypothesis also implied by the results of Section 6.5. The other distinguishing property of Paragraph Vectors is their very high correlation when regressing on the individual features.

#### 6.6.3 SKIPTHOUGHT VECTORS

It is hard to disentangle the properties of Skipthought vectors from the high dimensionality of the pretrained vectors we used. In general, Skipthought vectors performed poorly. They only performed better than other vector functions with one selector, Clustering, although their performance with this selector was significant.

## 7 CONCLUSIONS

We have identified differences in different forms of sentence vectors when applied to real-world tasks. In particular, each sentence vector form seems to be more successful when used in a particular way. Roughly speaking, Arora's vectors excel at judging the similarity of two sentences while Paragraph Vectors excel at representing document vectors, and at representing features as dimensions of vectors. While we do not have enough data to pinpoint the strengths of Skipthought vectors, they seem to work well in specific contexts that our work did not fully explore. These differences are extremely significant, and will likely make or break real-world applications. Therefore, special care should be taken when selecting the sentence vector method for a real-world task.

### ACKNOWLEDGMENTS

This material is based upon work supported by the National Science Foundation under Grant No. 1659788 and 1359275

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

## A    APPENDIX A

The first choice of the greedy algorithm is simple: it chooses the sentence with maximum cosine similarity to the document vector:

$$\bar{s}_1 = \underset{\bar{s} \in S}{\operatorname{argmax}} \, \bar{s} \cdot \bar{d}$$

(Recall that all vectors have unit-length, so cosine similarity is equivalent to the dot product).

To select the second vector, the greedy algorithm is maximizing the following equation:

$$\bar{s}_2 = \underset{\bar{s} \in S'}{\operatorname{argmax}} \left( \frac{\bar{s} + \bar{s}_1}{\|\bar{s} + \bar{s}_1\|} \right) \cdot \bar{d} \tag{3}$$

$$= \underset{\bar{s} \in S'}{\operatorname{argmax}} \, \frac{\bar{d} \cdot \bar{s}_1 + \bar{d} \cdot \bar{s}}{\sqrt{1 + \bar{s}_1 \cdot \bar{s}}} \tag{4}$$

$$\tag{5}$$

where S' is the set of remaining sentences [2]

In general, the behavior can be described by:

$$\bar{s}_{i+1} = \bar{s}_{total} \cdot \bar{d} \tag{6}$$

$$\bar{s}_{i+1} = \underset{\bar{s} \in S'}{\operatorname{argmax}} \, \frac{\frac{1}{i+1}\bar{s} + \frac{i}{i+1}\bar{s}_p}{\|\frac{1}{i+1}\bar{s} + \frac{i}{i+1}\bar{s}_p\|} \cdot \bar{d} \tag{7}$$

$$= \underset{\bar{s} \in S'}{\operatorname{argmax}} \, \frac{i(\bar{d} \cdot \bar{s}_p) + \bar{d} \cdot \bar{s}}{\sqrt{i^2 + 1 + 2i\bar{s}_p \cdot \bar{s}}} \tag{8}$$

$$\text{where } \bar{s}_p = \frac{\sum_{j=0}^{i} \bar{s}_j}{\|\sum_{j=0}^{i} \bar{s}_j\|}$$

---

[2]Note that the results reported above do not represent the Greedy algorithm averaging together the vectors, though the difference is minimal for SIF Average and Arora vectors(see Section 6.5 for more information)

# B  APPENDIX B

|  | SIF Average | Arora | Paragraph Vectors | Skipthought |
|---|---|---|---|---|
| **Near Nonredundant** | -1.98 | -1.72 | -0.734 | +3.81 |
| **Brute Force** | -0.323 | -0.256 | -1.24 | +3.42 |
| **Near-then-Redundancy** | +0.739 | -0.584 | -0.205 | +3.46 |
| **Max Similarity** | -0.323 | -0.256 | -1.24 | NA |
| **Greedy** | +0.254 | -0.813 | -3.11 | NA |
| **Near** | -0.868 | -0.0652 | -5.53 | +1.76 |
| **Total Average** | -.417 | -.614 | -2.01 | +2.74 |

Table 2: A comparison of document vector methods. Numbers represent the difference in ROUGE-1 scores between document vector methods. Positive numbers represent a gain when using *docvec-avg*. Selectors which do not use the document vector have been omitted.

