# OpenReview forum: "Exploring Sentence Vectors Through Automatic Summarization"
_ICLR.cc/2018/Conference — Reject_

### Official Review · AnonReviewer1 · 2017-11-24
**Standard extractive summarisation techniques using sentence vectors**

**Rating:** 2
**Confidence:** 5

**Review:**

The authors report a number of experiments using off-the-shelf sentence embedding methods for performing extractive summarisation, using a number of simple methods for choosing the extracted sentences. Unfortunately the contribution is too minor, and the work too incremental, to be worthy of a place at a top-tier international conference such as ICLR. The overall presentation is also below the required standard. The work would be better suited for a focused summarisation workshop, where there would be more interest from the participants.

Some of the statements motivating the work are questionable. I don't know if sentence vectors *in particular* have been especially successful in recent NLP (unless we count neural MT with attention as using "sentence vectors"). It's also not the case that the sentence reordering and text simplification problems have been solved, as is suggested on p.2.

The most effective method is a simple greedy technique. I'm not sure I'd describe this as being "based on fundamental principles of vector semantics" (p.4).

The citations often have the authors mentioned twice.

The reference to "making or breaking applications" in the conclusion strikes me as premature to say the least.

---

### Official Review · AnonReviewer3 · 2017-11-26
**Preliminary work, limited technical contribution**

**Rating:** 2
**Confidence:** 5

**Review:**

This paper explored the effectiveness of four existing sentence embedding models on ten different document summarization methods leveraging various works in the literature. Evaluation has been conducted on the DUC-2004 dataset and ROUGE-1 and ROUGE-2 scores are reported.

Overall, the paper significantly suffered from an immature writing style, numerous typos/grammatical mistakes, inconsistent organization of content, and importantly, limited technical contribution. Many recent sentence embedding models are missed such as those from Lin et al. (2017), Gan et al. (2017), Conneau et al. (2017), Jernite et al. (2017) etc. The evaluation and discussion sections were mostly unclear and the results of poorly performing methods were not reported at all making the comparisons and arguments difficult to comprehend.

In general, the paper seemed to be an ordinary reporting of some preliminary work, which at its current stage would not be much impactful to the research community.

---

### Official Review · AnonReviewer2 · 2017-11-27
**Evaluation and analysis of sentence embeddings for automatic summarization is good, but better justification of motivations and significance of results needed.**

**Rating:** 3
**Confidence:** 5

**Review:**

This paper examines a number of sentence and document embedding methods for automatic summarization. It pairs a number of recent sentence embedding algorithms (e.g., Paragraph Vectors and Skip-Thought Vectors) with several simple summarization decoding algorithms for sentence selection, and evaluates the resulting output summary on DUC 2004 using ROUGE, based on the general intuition that the selected summary should be similar to the original document in the vector space induced by the embedding algorithm. It further provides a number of analyses of the sentence representations as they relate to summarization, and other aspects of the summarization process including the decoding algorithm.

The paper was well written and easy to understand. I appreciate the effort to apply these representation techniques in an extrinsic task.

However, the signficance of the results may be limited, because the paper does not respond to a long line of work in summarization literature which have addressed many of the same points. In particular, I worry that the paper may in part be reinventing the wheel, in that many of the results are quite incremental with respect to previous observations in the field.

Greedy decoding and non-redundancy: many methods in summarization use greedy decoding algorithms. For example, SumBasic (Nenkova and Vanderwende, 2005), and HierSum (Haghighi and Vanderwende, 2009) are two such papers. This specific topic has been thoroughly expanded on by the work on greedy decoding for submodular objective functions in summarization (Lin and Bilmes, 2011), as well as many papers which focus on how to optimize for both informativeness and non-redundancy (Kulesza and Taskar, 2012).

The idea that the summary should be similar to the entire document is known as centrality. Some papers that exploit or examine that property include (Nenkova and Vanderwende, 2005; Louis and Nenkova, 2009; Cheung and Penn, 2013)

Another possible reading of the paper is that its novelty lies in the evaluation of sentence embedding models, specifically. However, these methods were not designed for summarization, and I don't see why they should necessarily work well for this task out of the box with simple decoding algorithms without finetuning. Also, the ROUGE results are so far from the SotA that I'm not sure what the value of analyzing this suite of techniques is.

In summary, I understand that this paper does not attempt to produce a state-of-the-art summarization system, but I find it hard to understand how it contributes to our understanding of future progress in the summmarization field. If the goal is to use summarization as an extrinsic evaluation of sentence embedding models, there needs to be better justification of this is a good idea when there are so many other issues in content selection that are not due to sentence embedding quality, but which affect summarization results.

References:

Nenkova and Vanderwende, 2005. The impact of frequency on summarization. Tech report.
Haghighi and Vanderwende, 2009. Exploring content models for multi-document summarization. NAACL-HLT 2009.
Lin and Bilmes, 2011. A class of submodular functions for document summarization. ACL-HLT 2011.
Kulesza and Taskar, 2012. Learning Determinantal Point Processes.
Louis and Nenkova, 2009. Automatically evaluating content selection in summarization without human models. EMNLP 2009.
Cheung and Penn, 2013. Towards Robust Abstractive Multi-Document Summarization: A Caseframe Analysis of Centrality and Domain. ACL 2013.

Other notes:
The acknowledgements seem to break double-blind reviewing.

---

### Decision · Program_Chairs · 2018-01-29
**ICLR 2018 Conference Acceptance Decision**

**Decision:**

Reject

**Comment:**

This work is interested in using sentence vector representations as a method for both doing extractive summarization and as a way to better understand the structure of vector representations. While the methodological aspects utilize representation learning, the reviewers felt that the main thrust of the work would be better suited for a summarization workshop or even NLP venue, as it did not target DL based contributions. Additionally they felt that the work did not significantly engage with the long literature on the problem of summarization.